# Ultrastrong and ductile steel welds achieved by fine interlocking microstructures with film-like retained austenite

Joonoh Moon [1,9] ✉, Gyuyeol Bae [2,9] ✉, Bo-Young Jeong[2], Chansun Shin[3], Min-Ji Kwon[1], Dong-Ik Kim [4], Dong-Jun Choi[4], Bong Ho Lee[5], Chang-Hoon Lee[6], Hyun-Uk Hong [1], Dong-Woo Suh[7] & Dirk Ponge[8]

The degradation of mechanical properties caused by grain coarsening or the formation of brittle phases during welding reduces the longevity of products. Here, we report advances in the weld quality of ultra-high strength steels by utilizing Nb and Cr instead of Ni. Sole addition of Cr, as an alternative to Ni, has limitations in developing fine weld microstructure, while it is revealed that the coupling effects of Nb and Cr additions make a finer interlocking weld microstructures with a higher fraction of retained austenite due to the decrease in austenite to acicular ferrite and bainite transformation temperature and carbon activity. As a result, an alloying design with Nb and Cr creates ultrastrong and ductile steel welds with enhanced tensile properties, impact toughness, and fatigue strength, at 45% lower material costs and lower environmental impact by removing Ni.

Discontinuities and the degradation of mechanical properties in welds decrease the integrity and life of metal products. For steels, as the most widely used material family, overcoming the formation of local brittle zones and cracks in welds is a long-standing challenge. Recently, as demands for the safety of structures and the weight reduction of structural parts are rising, the development and application of ultra-high strength steels (UHSS) with a tensile strength of more than 1 GPa are increasing. In particular, various kinds of UHSS such as transformation-induced plasticity (TRIP) steel, dual phase (DP) steel, and complex phase (CP) steel, have been developed for automotive applications[1–8]. To achieve an excellent balance of ultra-high strength and large ductility, the microstructures of automotive UHSS have been optimized via various combinations of hard phases (martensite and bainite) and ductile phases (ferrite and retained austenite). Irrespective of their excellent mechanical properties as base metals, significant degradation of mechanical properties, e.g., reduction of ductility, occurs in the welds[9–12], potentially causing various problems. Therefore, the challenge in welding of automotive UHSS is to improve strength without a loss in ductility (elongation and toughness) in the welds.

We make the welds stronger and more ductile with a weld microstructure design based on an alloy design concept that utilizes Nb and Cr to replace Ni. In the welds for UHSS, Ni has been conventionally used to obtain high ductility. However, excessive use of Ni increases the production cost and environmental burden. Many studies have reported the beneficial effects of Nb addition in steel products[13–17], i.e., Nb enhances the tensile properties, resistance to hydrogen embrittlement, and creep properties owing to its contribution to grain refinement and precipitation strengthening. During hot-rolling and heat treatment processes for the fabrication of steel

[1]Department of Materials Convergence and System Engineering, Changwon National University, Changwon, Gyeongnam, Republic of Korea. [2]Steel Solution Research Lab., Technical Research Lab., POSCO, Incheon, Republic of Korea. [3]Department of Materials Science and Engineering, Myongji University, Yongin, Republic of Korea. [4]Energy Materials Research Center, Korea Institute of Science and Technology, Seoul, Republic of Korea. [5]Advanced Analysis Team, Inst. of Next-Generation Semicond. Convergence Technol., Daegu Gyeongbuk Institute of Science and Technology, Daegu, Republic of Korea. [6]Steel Department, Korea Institute of Materials Science, Changwon, Republic of Korea. [7]Graduate Institute of Ferrous & Energy Materials Technology, Pohang University of Science and Technology, Pohang, Gyeongbuk, Republic of Korea. [8]Max-Planck-Institut für Eisenforschung, Düsseldorf, Germany. [9]These authors jointly supervised this work: Joonoh Moon, Gyuyeol Bae. ✉e-mail: mjo99@changwon.ac.kr; gbae@posco.com

products, solute Nb atoms and fine NbC particles inhibit the migration of austenite grain boundaries, refining the grain size. In addition, Nb segregation at austenite grain boundaries stabilizes them and thus delays the austenite to ferrite transformation, ultimately promoting the formation of non-polygonal ferrite and bainite. Solute Nb atoms in austenite matrix also increase the hardenability of steels[18]. The grain refinement and the formation of bainitic structure by Nb addition instead of Ni can improve both strength and ductility. However, the effect of Nb alloying in welds is controversial[19–21]. Nb addition can also lead to grain refinement in the welds, improving mechanical properties. On the other hand, solute Nb increases the hardenability of the welds and then forms brittle phases during fast cooling, deteriorating the ductility in the welds. In addition, coarse NbC particles precipitated in reheated welds during multi-pass welding degrade the impact toughness. These aspects have posed a critical hurdle limiting the use of Nb in UHSS welds. Next, Cr also significantly improves the hardenability of steels[22] and its effect is higher than Ni. In addition, the small addition of Cr increases the stability of retained austenite, finally improving mechanical properties[23].

Here, we report a weld microstructure design for 1.0 GPa- and 1.2 GPa-grade UHSS with excellent mechanical properties, reduced material costs, and lower environmental impact. To achieve a balance of high strength and ductility in the welds, we aims to make a fine interlocking microstructure consisting of acicular ferrite and bainite with film-like retained austenite. To this end, we investigate the microstructures in welds when Cr is added alone instead of Ni and Cr + Nb are added simultaneously, finally find that a complex addition of Nb and Cr shows a better performance with a desirable microstructure. When Cr is added alone instead of Ni, the fraction of retained austenite slightly increases in the welds but the effective grain size also increases, finally fails to obtain an optimized microstructure, i.e., a sole addition of Cr decoupled from Nb has limitation in compensating the reduction of hardenability caused by removing high amounts of Ni (1.0–1.5%). For these reasons, we finally decide to add both Nb and Cr instead of Ni in the welds of UHSS. With this alloy design, the material costs are reduced by approximately 45%. In addition, both tensile strengths and uniform elongations in the welds of UHSS increase by 14 MPa and 0.6% (for 1.0 GPa-grade UHSS welds) and 86 MPa and 0.1% (for 1.2 GPa-grade UHSS welds). The impact and fatigue properties are also improved significantly. Impact tests are performed at a temperature range of −80 °C to 0 °C and the results show that impact toughness increases by at least 13% (for 1.0 GPa-grade UHSS welds) in all test temperatures, and fatigue limit increases by 240% (for 1.0 GPa-grade UHSS welds).

## Results and discussion

Two kinds of conventional welds containing high Ni contents were selected as starting points for exploring 1.0 GPa- and 1.2 GPa-grade UHSS steel welds. Their compositions in the welds are Fe-0.08C-0.72Si-1.83Mn-1.44Ni-0.010Nb-0.34Cr-Ti-Mo (wt%) (1.0 GPa H-Ni steel weld, see Table 1) and Fe-0.11C-0.37Si-2.2Mn-1.13Ni-0.016Nb-0.34Cr-Ti-Mo

(wt%) (1.2 GPa H-Ni steel weld, see Table 1), respectively. Ni increases the hardenability, promoting the formation of bainite in the welds[24,25]. Additionally, Ni, as a strong austenite stabilizing element, aids the formation of retained austenite, improving ductility[25,26]. However, excessive use of Ni increases product costs and is considered an environmental burden[27]. Therefore, we searched for a weld composition design for 1.0 GPa- and 1.2 GPa-grade UHSS and found that the addition of small amounts of Nb and Cr can effectively replace Ni. The final developed compositions are based on Fe-0.08C-0.59Si-1.85Mn-0.038Nb-0.73Cr-Ti-Mo for the 1.0 GPa UHSS weld (1.0 GPa H-Nb steel weld, see Table 1) and Fe-0.12C-0.28Si-2.3Mn-0.035Nb-0.64Cr-Ti-Mo for the 1.2 GPa UHSS weld (1.2 GPa H-Nb steel weld, see Table 1). In addition, we reduced the material costs by approximately 45% through the replacement of Ni by Nb and Cr, as shown in Table 1. It should be noted that we also evaluated the role of Cr addition decoupled from the effect of Nb on the weld microstructure and properties, and finally failed to obtain the desirable weld structure. The results are discussed below.

Figure 1a shows a schematic of the hierarchical microstructure formed in the single-pass welded joint of UHSS. During single-pass welding, the molten metal solidifies into δ-ferrite first, and δ-ferrite then transforms into austenite. Next, the austenite subsequently transforms into various kinds of phases, depending on the chemical compositions in the welds. After solidification, the acicular ferrite plates nucleate at the interface between the austenite and inclusions and then grow radially, rendering the interlocking structure. The bainite phases nucleate at the prior austenite grain boundaries (PAGB) and grow into austenite. During this bainite transformation, the C atoms, a strong austenite stabilizer, precipitate cementite with the formula $Fe_3C$ along the lath boundaries or within the laths. Then, the addition of certain elements, e.g., Si, reduces the stability of cementite, promoting the formation of retained austenite at the lath boundaries. Finally, the remaining austenite matrix is transformed to martensite. Figure 1b, c show microstructures in the 1.2 GPa H-Nb weld. As shown in Fig. 1a, acicular ferrite plates formed around the inclusions, and these nucleants were then identified as Al-rich oxides (Fig. 1e). The EBSD inverse pole figure (IPF) map in Fig. 1d indicates that the acicular ferrite has a very complex structure with highly misoriented grain boundaries, macroscopically exhibiting the interlocking microstructure. Figure 1c is the TEM micrograph showing the bainite and retained austenite phases in the welds.

Figure 2 shows the effects of small amounts of Nb and Cr additions instead of Ni on the effective grain size of the interlocking microstructure and the fraction of retained austenite in the welds. Figure 2a, c are EBSD IPF maps of the 1.0 GPa H-Ni and 1.0 GPa H-Nb welds, respectively, and indicate that the H-Nb welds have a finer interlocking microstructure than the H-Ni welds. Figure 2b, d are EBSD phase maps showing the distribution of retained austenite in the 1.0 GPa H-Ni and 1.0 GPa H-Nb welds, respectively, indicating that the fraction of retained austenite increased significantly with the addition of Nb and Cr. To characterize the retained austenite and its shape

**Table 1 | Chemical compositions of the welds**

| GIGA steel weld | Chemical composition, wt.% | | | | | | | | Alloy cost (USD/ton) |
|---|---|---|---|---|---|---|---|---|---|
| | C | Si | Mn | Al | Ni | Nb | Cr | Ti, Mo | |
| 1.2 GPa H-Nb <br> *(1.2 GPa H-Nb/L-Ni)* | 0.12 | 0.28 | 2.3 | 0.033 | 0.01 | 0.035 | 0.64 | added | 243 |
| 1.2 GPa H-Ni <br> *(1.2 GPa H-Ni/L-Nb)* | 0.11 | 0.37 | 2.2 | 0.039 | 1.13 | 0.016 | 0.30 | | 458 |
| 1.0 GPa H-Nb <br> *(1.0 GPa H-Nb/L-Ni)* | 0.08 | 0.59 | 1.85 | 0.016 | 0.01 | 0.038 | 0.73 | | 302 |
| 1.0 GPa H-Ni <br> *(1.0 GPa H-Ni/L-Nb)* | 0.08 | 0.72 | 1.83 | 0.013 | 1.44 | 0.010 | 0.34 | | 540 |

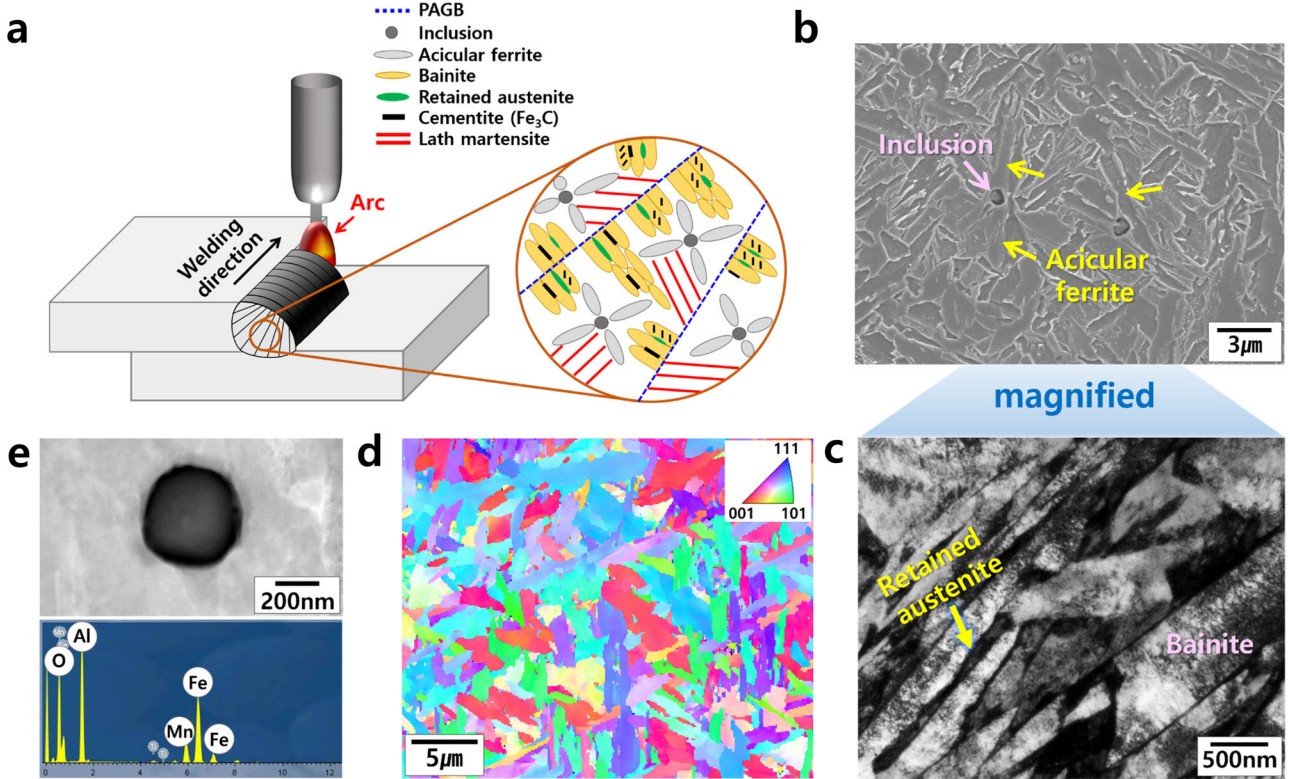

**Fig. 1 | Representative microstructural aspects of the ultrastrong and ductile steel weld. a** Schematic of hierarchical microstructures in the weld. **b**–**d** SEM and TEM images and EBSD IPF map of the 1.2 GPa H-Nb steel weld, respectively. **e** Inclusion acting as a nucleation site for acicular ferrite. PAGB in Fig. 1a denotes a prior austenite grain boundary.

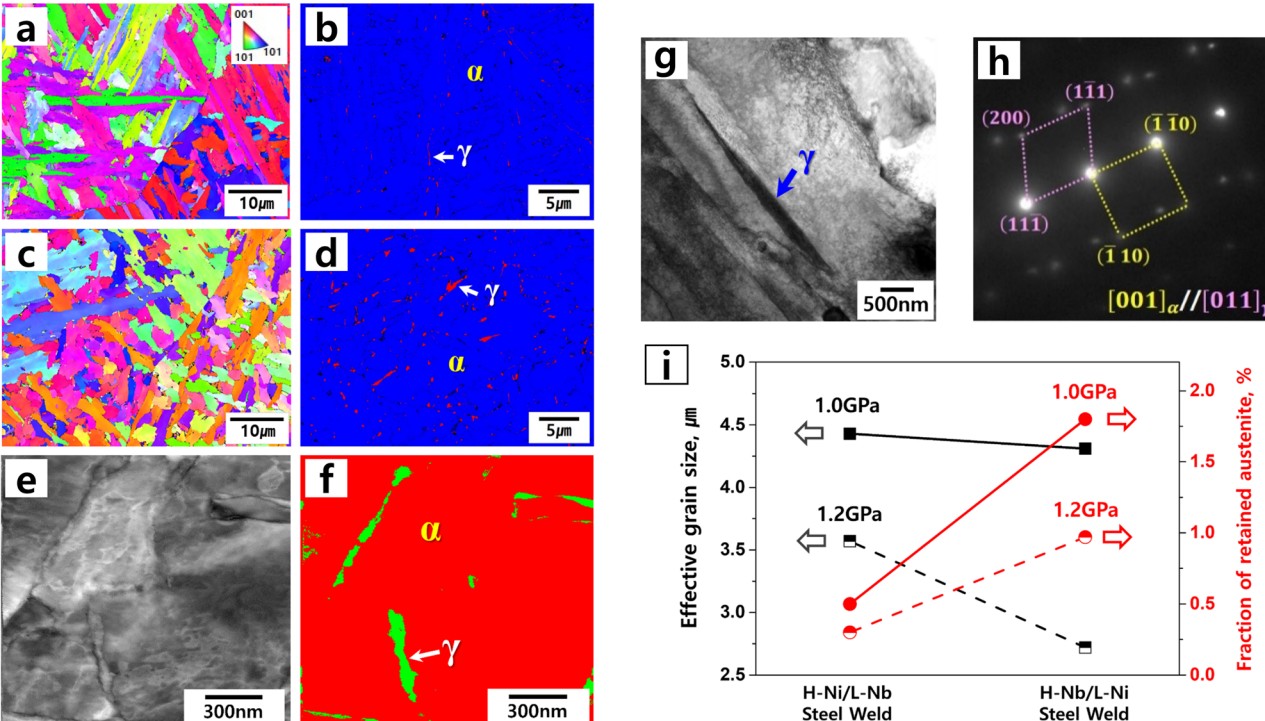

**Fig. 2 | Hierarchical architecture of the ultrastrong and ductile steel welds. a**, **b** EBSD IPF and phase map of the 1.0 GPa H-Ni steel weld. **c**, **d** EBSD IPF and phase map of the 1.0 GPa H-Nb steel weld. **e**, **f** TEM image and phase map of **d**. **g**, **h** STEM image and SADP analyses for film-like retained austenite of **d**. **i** Effective grain size and fraction of retained austenite in the welds measured from the EBSD analyses.

**Table 2 | Effective grain size and fraction of retained austenite in the welds with different contents (wt%) of alloying elements**

| GIGA steel weld | | Alloying elements, wt% | | | Effective grain size, µm | Fraction of retained austenite, % |
|---|---|---|---|---|---|---|
| | | Ni | Nb | Cr | | |
| 1.2GPa L-Nb (1.2GPa L-Nb/L-Ni) | Reference * | 0.01 | 0.016 | 0.73 | 4.5 | 0.4 |
| 1.2 GPa H-Nb (1.2 GPa H-Nb/L-Ni) | Developed | 0.01 | 0.035 | 0.64 | 2.7 | 1.0 |
| 1.2 GPa H-Ni (1.2 GPa H-Ni/L-Nb) | Conventional | 1.13 | 0.016 | 0.34 | 3.6 | 0.3 |
| 1.0GPa L-Nb (1.0GPa L-Nb/L-Ni) | Reference * | 0.01 | 0.011 | 0.70 | 5.2 | 0.9 |
| 1.0 GPa H-Nb (1.0 GPa H-Nb/L-Ni) | Developed | 0.01 | 0.038 | 0.73 | 4.3 | 1.8 |
| 1.0 GPa H-Ni (1.0 GPa H-Ni/L-Nb) | Conventional | 1.44 | 0.010 | 0.34 | 4.4 | 0.5 |

\* Reference: Sole addition of Cr decoupled from Nb instead of Ni.

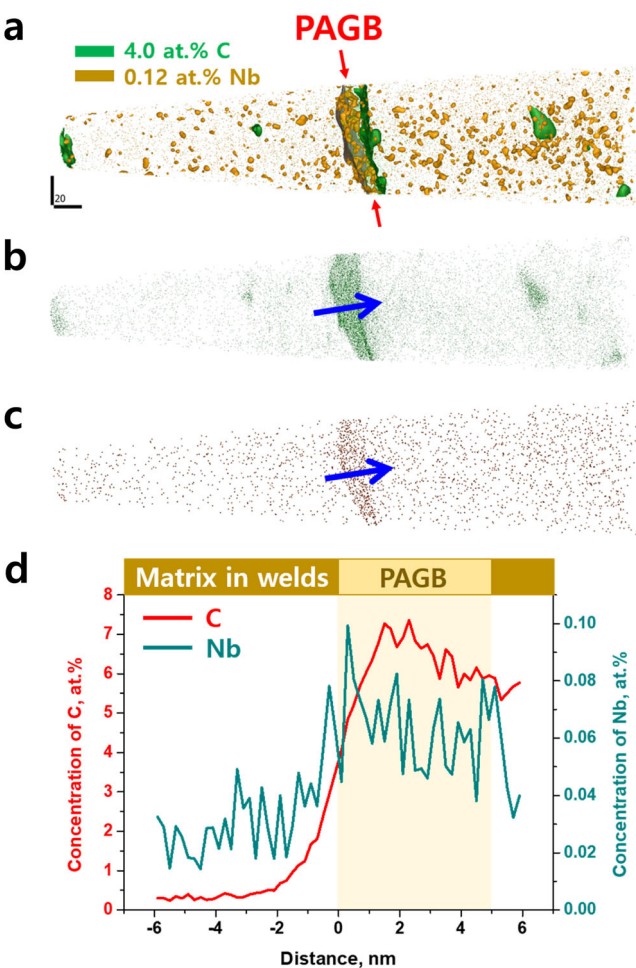

**Fig. 3 | Results of APT analyses representing Nb and C segregation along the PAGB in the weld centerline of the 1.0 GPa H-Nb steel weld. a** Iso-concentration surface image. **b, c** C and Nb atom maps, respectively. **d** Compositional profiles of C and Nb across the PAGB interface (measured along the direction marked by blue arrows in b and c. Distance '0' indicates the point that PAGB starts while following in the direction of the blue arrow).

precisely, we performed TEM analyses using ASTAR (orientation and phase mapping tool on TEM). Figure 2f shows the TEM phase map corresponding to the TEM image of Fig. 2e, clearly revealing that film-like retained austenite was distributed along the bainite lath

boundaries. Finally, we identified the retained austenite via selected area diffraction pattern (SADP) analyses, as shown in Fig. 2g, h. Figure 2i summarizes the results measured from the EBSD analyses, i.e., in both 1.0 GPa and 1.2 GPa steel welds, the effective grain size of the interlocking microstructure decreased, and the fraction of film-like retained austenite increased with the addition of small amounts of Nb and Cr by replacing Ni. Compared to the conventional H-Ni steel welds, the developed H-Nb steel welds have more than three times the fraction of retained austenite (3.3 times in 1.2 GPa steel weld and 3.6 times in 1.0 GPa steel weld, as shown in Table 2). Meanwhile, Table 2 shows that a sole addition of Cr decoupled from Nb instead of Ni slightly increased the fraction of retained austenite but did not make the weld finer, indicating that a single addition of Cr without Nb is not enough to compensate the reduction of hardenability caused by removing high amounts of Ni.

Therefore, the refinement of the interlocking microstructure in H-Nb steel welds is closely related to the coupling effects of Nb and Cr additions, originated from the segregation of Nb atoms along the PAGB and the enhancement of hardenability by increasing the solute Nb and Cr contents. According to a previous report[18], solute Nb atoms segregate to the PAGB during the welding thermal cycle due to the large lattice misfit of Nb atoms in Fe and then stabilize them, delaying the austenite to bainite transformation to lower temperatures and thereby leading to the formation of fine bainitic structures. In this context, we characterized the atomic-scale elemental compositions across the PAGB in the welds by atom probe tomography (APT) analyses. Figure 3a presents 0.12 at% Nb and 4.0 at% C iso-concentration surface images in 1.0 GPa H-Nb welds, and Fig. 3b, c are Nb and C atom maps, respectively. These results demonstrate the segregation of solute C and Nb atoms along the PAGB, and the extent of their enrichment was analyzed by the composition profiles across the PAGB, as shown in Fig. 3d. In addition, the coupling effect demonstrated by a solid solution of Nb and Cr atoms is more greater than the effect of a solid solution of Ni atoms in increasing hardenability[18–22], indicating that Nb and Cr additions instead of Ni lead to decrease in the transformation temperature of austenite to acicular ferrite and bainite. From these facts, the finer interlocking microstructure in the H-Nb steel welds compared to the H-Ni steel welds, as shown in Fig. 2i, can be understood as the effect of Nb and Cr additions, i.e., the austenite to acicular ferrite and bainite transformation was suppressed by the segregation of solute Nb atoms along the PAGB and the increased hardenability, resulting in the formation of acicular ferrite plates and bainite laths at lower temperatures, ultimately rendering a fine interlocking microstructure. To confirm the variation in the austenite to acicular ferrite and bainite transformation temperature with the addition of Nb and Cr, a continuous cooling transformation (CCT)

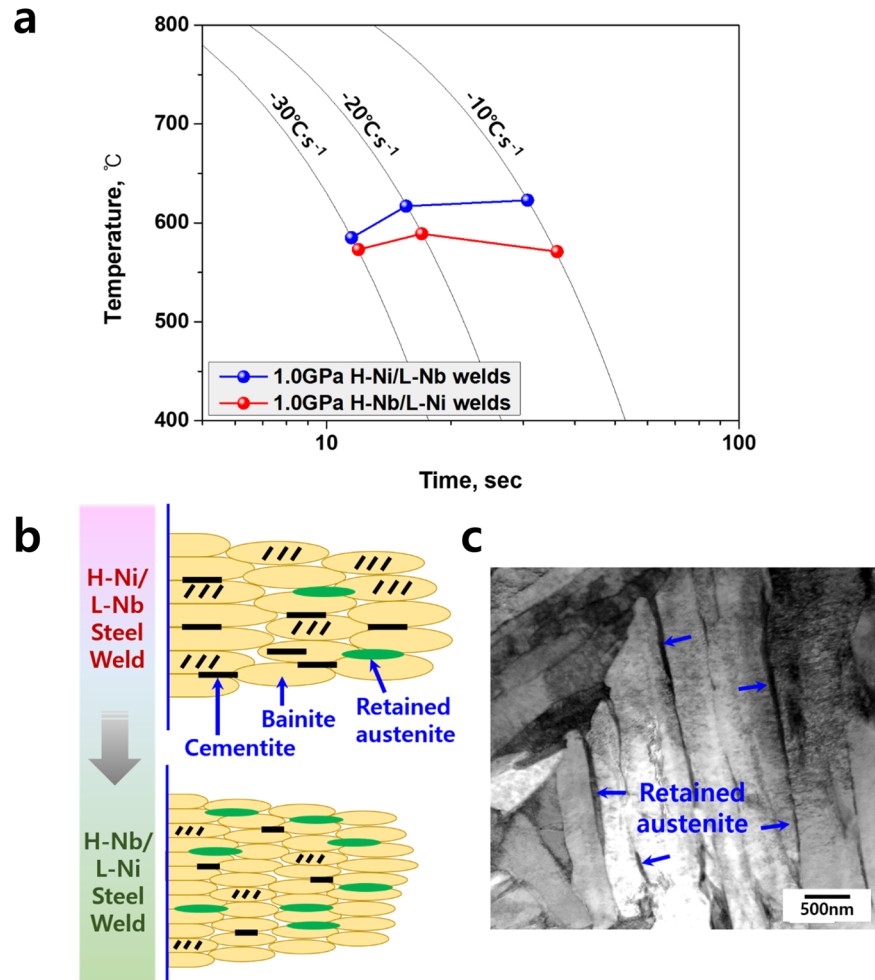

**Fig. 4 | Correlation between the phase transformation temperature and film-like retained austenite formation. a** CCT diagram showing the start temperatures of austenite to ferrite transformation in the welds. **b** Schematic of bainite transformation characteristics in the UHSS welds. **c** Representative TEM image revealing the film-like retained austenite between the lath boundaries of carbide-free bainite in the 1.0 GPa H-Nb steel weld.

diagram was drawn through dilatometer experiments (Fig. 4a), and the results show that the start temperatures for the austenite to bainite transformation decreased with the addition of Nb and Cr, indicating that a small amount of Nb and Cr greatly improved the hardenability in the welds despite the elimination of a large amount of Ni.

Compared to conventional H-Ni welds, one important microstructural feature in H-Nb welds is a higher fraction of retained austenite, as shown in Fig. 2. We can also understand this result as the effect of the decrease in bainite transformation temperature as a consequence of adding Nb and Cr instead of Ni. During the upper bainite transformation stage, C atoms are ejected from the bainite laths and then enriched at the lath boundaries, ultimately resulting in interlath cementite. In this context, the decrease in transformation temperature caused by adding Nb makes the formation of cementite during upper bainite transformation kinetically difficult due to insufficient time for precipitation. Due to the additional C partitioning to the untransformed austenite between the bainite laths, this austenite becomes more stable. This leads ultimately to a relatively high fraction of retained austenite in the welds. In addition, the fraction of retained austenite in Fig. 2i was also affected by the activity of carbon atoms[28]. Leslie and Rauch investigated the effects of alloying elements on the precipitation of cementite in low-carbon steels and reported that a decrease in the activity of carbon increases the solubility of carbon in the ferrite matrix relative to cementite, ultimately suppressing the precipitation of cementite. Here, we confirm the effects of the addition

of small amounts of Nb and Cr instead of Ni on the activity of carbon through thermodynamic calculations. For this purpose, we calculated the changes in carbon activity with increasing Nb, Cr, and Ni in the same composition with 1.0 GPa steel welds using Thermo-Calc. software (TCFE 12 database). The calculation results, shown in Supplementary Fig. 1, show that the activity of carbon decreases with the addition of Nb and Cr, while it increases with the addition of Ni, indicating that the addition of Nb and Cr instead of Ni increases the carbon dissolved in the matrix and thereby promote the formation of retained austenite in the welds despite the elimination of Ni. In Supplementary Fig. 1, we can also see that Nb is more effective in reducing the activity of carbon than Cr, i.e., Nb has a similar effect despite the addition of a smaller amount compared to Cr. Our observations in Fig. 2 are in quite good agreement with the calculations. In this regard, Fig. 4b shows a schematic of the effect of small amounts of Nb and Cr additions on the formation of bainitic structures in the welds, i.e., the precipitation of cementite is suppressed and the formation of retained austenite increases when Nb and Cr are added instead of Ni. Figure 4c presents a representative bainite microstructure in the 1.0 GPa H-Nb steel weld showing carbide-free bainite with film-like retained austenite.

To evaluate the mechanical properties of the welds, we carried out micro-tensile tests, in-situ EBSD tensile tests, Charpy V-notch impact tests, and fatigue tests. For the micro-tensile tests, we detached the tensile test specimens along the welding direction, as shown in Supplementary Fig. 2a. Supplementary Fig. 2b presents the fractured

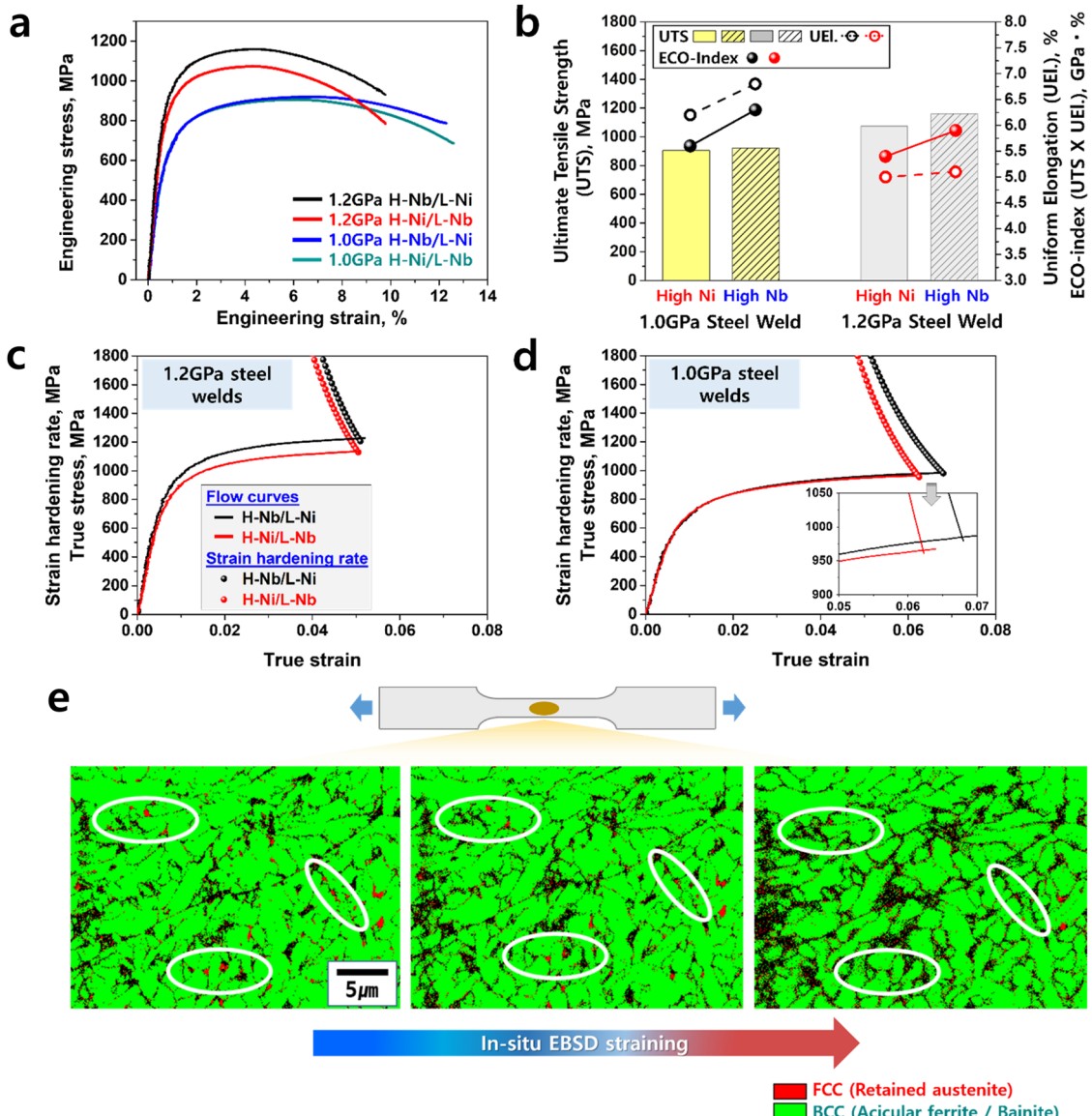

**Fig. 5 | Micro-tensile/in-situ EBSD tensile test results for the ultrastrong and ductile steel welds. a** Engineering stress–strain curves. **b** Tensile strength, uniform elongation, and ECO-index values of the welds based on the micro-tensile test results. **c**, **d** True stress–strain curves with strain hardening rates of 1.2 GPa and 1.0 GPa steel welds, respectively. **e** Variations in the EBSD phase maps with increasing strain during the in-situ EBSD tensile tests.

samples after the micro-tensile tests. It should be noted that previous works[29–32] reported that the ratio of t/d (where t represents the thickness of the flat tensile specimen and d signifies the grain size) should exceed 10 to guarantee that the mechanical properties measured in micro-tensile testing are representative of the material's bulk behavior. Recent theoretical study[33] have shown that at the values greater than 15, the yield strength becomes size-independent and converges toward the bulk yield strength. In this study, the value of t/d is 113–185, significantly exceeding the proposed critical threshold of 10 or 15. Therefore, the mechanical properties derived from the sub-size tensile specimens utilized in this investigation can be confidently regarded as equivalent to the bulk material's properties. The results of the micro-tensile tests are shown in Fig. 5a–d. Figure 5a shows the engineering stress–strain curves of the welds. Compared to the conventional H-Ni steel welds, the developed H-Nb steel welds exhibit higher strength without loss of ductility. Figure 5c, d present the true stress–strain curves with strain hardening rates (generated through nonlinear fitting of experimental data) in 1.2 GPa and 1.0 GPa steel welds. The results show that H-Nb steel welds have higher strength and strain hardening

rates than H-Ni steel welds, leading to higher uniform elongation. In Fig. 5c, d, the strain values corresponding to the intersection points between the flow curves and strain hardening rate indicate uniform elongations. As a result, H-Nb steel welds have higher ECO-index values, which are the product of tensile strength and uniform elongation. A higher ECO index means better balance of strength and ductility. These excellent tensile properties of the developed H-Nb steel welds resulted from the combined effects of the fine interlocking microstructure and higher fraction of film-like retained austenite, as shown in Fig. 2. During tensile testing, wavy and complex microstructures make the pathway of cracks more tortuous than straight and simple microstructures until the specimen is fractured, increasing the strength[34,35]. In this context, the refinement of interlocking acicular ferrite and bainitic plates by adding Nb in H-Nb steel welds renders crack propagation more complicated. The finer structure also enhances the strength. Next, the stability of film-like retained austenite found in welds of this study is likely to be higher than that of blocky retained austenite and thus hinders crack propagation effectively, improving ductility[36]. In addition, the retained austenite absorbs initial strain

during tensile deformation, leading to an increase in ductility by the transformation-induced plasticity (TRIP) effect via the transformation of austenite to martensite[37]. Previous works[38,39] reported that a decrease in the fraction of retained austenite increases carbon content in retained austenite and thereby makes retained austenite too stable to be transformed to martensite during deformation, indicating that the effect of retained austenite on ductility improvement might be reduced by decreasing the fraction of metastable retained austenite. In Fig. 2i, it seems to be that the fraction of retained austenite in welds is quite small (~ 2%) to display TRIP effect during deformation compared to conventional TRIP steels. Nevertheless, the result of the in-situ EBSD tensile test in Fig. 5e shows clear evidence of the retained austenite to martensite transformation in welds during tensile deformation even though the fraction of retained austenite in welds was not as high as those in conventional TRIP steels, indicating that small increase in retained austenite fraction in welds by adding Nb and Cr instead of Ni had an effect on the improvement of ductility of welds, as clearly presented in Fig. 5b–d. Meanwhile, EBSD in Fig. 5e can resolve only the larger blocky retained austenite island but not the very fine retained austenite interlath films visible by TEM in Fig. 4c. With straining, the blocky retained austenite is gradually transformed to martensite, contributing to strain hardening and ductility improvement. Finally, the results of Charpy V-notch impact and fatigue tests are presented in Fig. 6. Both the fractured surfaces of H-Nb and H-Ni steel welds after

impact tests show a ductile dimple structures (Fig. 6b), but H-Nb steel welds have a higher toughness at all temperatures (Fig. 6a). The fatigue strengths are also higher in H-Nb steel welds than in conventional H-Ni steel welds (Fig. 6c). These great improvements in the impact and fatigue properties of the H-Nb steel welds are related to the refinement of the interlocking microstructure with a high fraction of film-like retained austenite. That is, an extension of the pathway for crack propagation by refining the hierarchical interlocking microstructure and an interruption of crack propagation by ductile retained austenite films contribute to much better impact toughness and fatigue resistance properties. Moreover, a decrease in bainite transformation temperature resulting from the addition of Nb and Cr might also contribute to the enhancement of fatigue properties in H-Nb steel welds. Garcia-Mateo et al.[40] revealed that a decrease in the bainite transformation temperature resulted in higher levels of carbon supersaturation in bainitic ferrite, ultimately leading to an increase in the c/a ratio in bainitic ferrite, i.e., the tetragonality of bainite increases with a decrease in the bainite transformation temperature. As shown in Fig. 4a and Supplementary Fig. 1, the addition of Nb and Cr instead of Ni decreased the bainite transformation temperature and carbon activity. These factors might increase the supersaturated carbon content in bainite and thereby increase the tetragonality of bainite. Additionally, it should be noted that carbon atoms ejected from acicular ferrite during the austenite to acicular ferrite transformation are

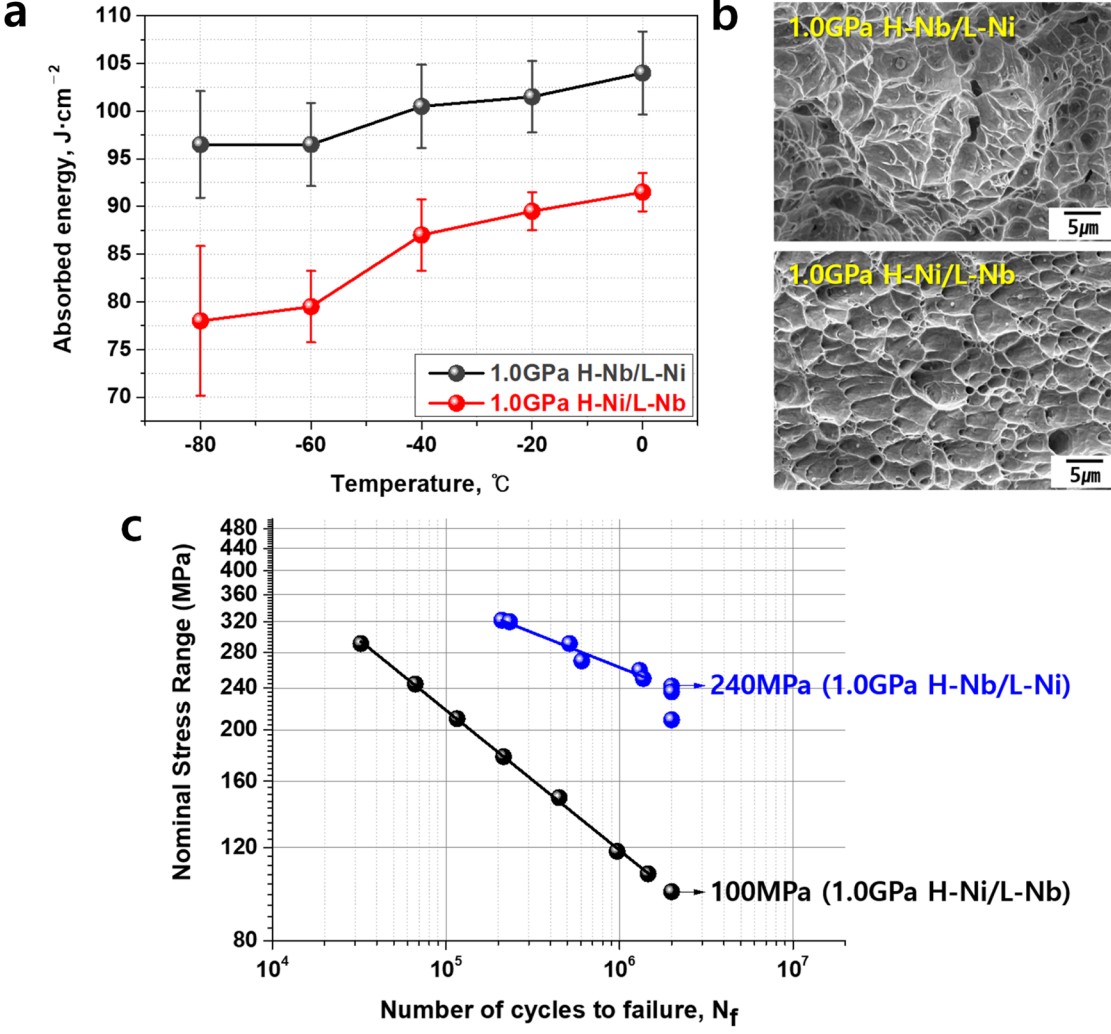

**Fig. 6 | Results of Charpy V-notch impact and fatigue tests for the ultrastrong and ductile steel welds. a** Absorbed energy values after impact tests. **b** SEM micrographs showing the fractured surface in the 1.0 GPa steel welds after the impact test at −40 °C. **c** Fatigue strength curves. All the error bars in **a** represent the standard deviation (*n* = 5 independent experiments).

enriched in austenite, enabling to play a role in increasing the tetragonality. Supplementary Fig. 3 shows the residual stresses measured in the 1.0 GPa H-Ni and 1.0 GPa H-Nb steel welds, indicating that tensile and compressive residual stresses were produced in the H-Ni and H-Nb steel welds, respectively. This different aspect of residual stress in H-Ni and H-Nb steel welds might result from the difference in their tetragonality, i.e., compared to H-Ni steel welds, a higher tetragonality of bainite formed in H-Nb steel welds might cause a compressive residual stress. D. Fukui et al[41]. reported that an internal residual stresses were originated from bain strain and tetragonality of martensite. Generally, tensile residual stress reduces the fatigue life by increasing fatigue crack growth, while compressive residual stress improves the fatigue properties by retarding crack propagation[42]. Therefore, it is believed that an increase in the tetragonality of bainite in H-Nb steel welds as a result of decreasing the transformation temperature and carbon activity would have contributed to the improvement in the fatigue properties, as shown in Fig. 6c.

In summary, we designed a weld microstructure for UHSS with better tensile properties, impact toughness, and fatigue resistance, lower product cost, and a lower environmental burden to compete with conventional welds. The compositions of the developed welds are based on Fe-0.08C-0.59Si-1.85Mn-0.038Nb-0.73Cr-Ti-Mo for the 1.0 GPa UHSS welds and Fe-0.12C-0.28Si-2.3Mn-0.035Nb-0.64Cr-Ti-Mo for the 1.2 GPa UHSS welds. Alloying with Nb and Cr renders the interlocking acicular ferrite and bainitic structures in the welds finer by stabilizing the PAGB and lowering the austenite to acicular ferrite and bainite transformation temperature. The formation of film-like retained austenite between bainite laths is promoted by adding Nb and Cr instead of Ni, thereby suppressing the precipitation of cementite. Therefore, the developed design of weld compositions and microstructures could be a potential strategy to replace conventional ones for welding UHSS owing to their better mechanical properties, cheaper material cost, and lower environmental impact.

## Methods

### Sample preparation

Here, we developed UHSS welds (1.0 GPa H-Nb/L-Ni steel weld: Fe-0.08C-0.59Si-1.85Mn-0.038Nb-0.73Cr-Ti-Mo and 1.2 GPa H-Nb/L-Ni steel weld: Fe-0.12C-0.28Si-2.3Mn-0.035Nb-0.64Cr-Ti-Mo, see Table 1) and compared them to conventional welds (1.0 GPa H-Ni/L-Nb weld: Fe-0.08C-0.72Si-1.83Mn-1.44Ni-0.010Nb-0.34Cr-Ti-Mo and 1.2 GPa H-Ni/L-Nb weld: Fe-0.11C-0.37Si-2.2Mn-1.13Ni-0.016Nb-0.34Cr-Ti-Mo, see Table 1). To prepare the welds, UHSS plates were machined with dimensions of 200 mm in width and 150 mm in length and lap-joint (10 mm in overlap length) metal active gas (MAG) welding was then performed on plates of 1.0 GPa- (2.0 mm in thickness and welding heat input of 2.3 kJ cm$^{-1}$) and 1.2 GPa-grade (1.6 mm in thickness and welding heat input of 1.7 kJ·cm$^{-1}$) UHSS, with a shielding gas of Ar+20% $CO_2$. Detailed welding conditions are presented in Supplementary Table 1.

### Microstructural characterization

The microstructures of the welds were observed with scanning electron microscopy (SEM), electron backscattered diffraction (EBSD), and transmission electron microscopy (TEM) analyses. Specimens for SEM observation were prepared by mechanical polishing and chemical etching in a mixed solution of 95 ml ethanol and 5 ml nitric acid. Thin foil specimens for the TEM analyses were prepared by twin-jet electrolytic polishing at 19 V and 200 mA with a mixed solution of 120 ml perchloric acid and 1060 ml methanol at -29 °C.

The segregation of alloying elements across the austenite grain boundary was analyzed using atom probe tomography (APT) with a CAMECA LEAP 4000X HR. Tip samples for the analyses were machined using a focused ion beam (FIB) milling. The tips were held in chamber with a vacuum of $1.0 \times 10^{-11}$ Torr at -233.15 °C (40 K), after which they were field-evaporated at an evaporation rate of 1.5% with 355 nm UV laser at a laser power of 100 pJ and a laser pulse rate of 200 kHz. The APT data were analyzed using Interactive Visualization and Analysis Software (AP Suite 6.1) of CAMECA instruments. The proximity histograms across the austenite grain boundary were analyzed using the standard analysis procedure of IVAS software.

### Mechanical testing

To observe the microstructure evolution during deformation, in-situ tensile experiments were performed using a scanning electron microscope (SEM, HITACHI S-4300 SE) equipped with EBSD (Bruker ESPRIT) and a tensile test attachment (Kammrath & Weiss MZT.M). The specimens were machined into a gauge length of 15 mm and a width of 4 mm. Prior to the tests, the specimens were mechanically polished and then electro-polished using a mixed solution of 73.14% ethanol, 10.02% 2n-butoxy, 7.8% perchloric acid, and 9.017% distilled water. In-situ EBSD straining tests were carried out with a crosshead speed of 5 µm·s$^{-1}$ and EBSD data were then acquired during in-situ deformation. Acquired EBSD data were analyzed using data post-processing software (TSL OIM 7.31).

The tensile properties of the welds were evaluated through micro-tensile testing using a Zwick Roell Z005 TN ProLine machine (Zwick-Roell, Ulm, Germany) with a 5-kN load cell. The specimens were machined within the welds along the direction of welding (Fig. 5a), with dimensions of 0.5 mm in thickness, 1 mm in width, and 5 mm in length. Tensile loading was applied to the specimen at a crosshead speed of 0.15 mm min$^{-1}$ (corresponding to a strain rate of $5 \times 10^{-4}$ s$^{-1}$). To measure strain during the test, digital image correlation (DIC) was employed. The specimen's surface was coated in white and an artificial quasi-stochastic color spray was applied. The GOM ARAMIS syste m was used to capture the deformation of the patterns on the surface of the specimen.

The impact toughness properties of the welds were evaluated through Charpy V-notch (CVN) testing using a CI-800-D machine (Tokyo Testing Machine, Japan) with an 800 J capacity. The specimens were prepared as 1.0 GPa-grade UHSS (2.9 mm in thickness) welds through butt-joint MAG welding resulting in full penetration welds and subsequently machined as sub-size V-notch specimens (2.5 mm in thickness) according to the JIS Z 2242 standard. The V-notch was formed in the center of the weld, and the CVN tests for the H-Nb/L-Ni and H-Ni/L-Nb steel welds were performed at −80 °C, −60 °C, −40 °C, −20 °C, and 0 °C. Three specimens were evaluated at each temperature, and the measured values were then converted to those corresponding to full-size specimens, as shown in Fig. 6a.

The fatigue properties of the welds were evaluated through bending fatigue testing using a PBF-60 machine (Tokyo Testing Machine, Japan) with a frequency of 15 Hz. The specimens were prepared as 1.0 GPa-grade UHSS (2.0 mm in thickness) welds through lap-joint MAG welding as described in the above section (see 'Sample preparation') and subsequently machined as fatigue specimens according to the JIS Z 2275 standard. The fatigue tests for the H-Nb/L-Ni and H-Ni/L-Nb steel welds were performed at each load with an R value (defined as Min. load/Max. load) of -1 and then plotted as shown in Fig. 6c.

### Measurement of residual stresses in the welds

The residual stresses of the welds were measured through X-ray diffractometer testing using an XSTRESS 3000 machine (Stresstech, Germany) with a Cr-tube (30 kV/6.7 mA) X-ray source and dual position sensitive MOS X-ray detectors. The specimens were prepared as 1.0 GPa-grade UHSS (2.9 mm in thickness) welds through butt-joint MAG welding, resulting in full penetration welds. The X-ray tube tip was positioned 1.0 mm (start point) away from the periphery of the weld, and subsequently, diffractometer measurements were carried out for H-Nb/L-Ni and H-Ni/L-Nb steel welds with a step of 1.0 mm (perpendicular direction to the welds) from the start point (i.e., 2.0 mm, 3.0 mm, 4.0 mm, 5.0 mm, respectively). The residual stresses for the

welds were estimated by measuring the inter-planar spacing (i.e., lattice distances) according to the 2θ values based on Bragg's law, as shown in Supplementary Fig. 3. It should be noted that the present datasets shown in Fig. 3 were measured from the center part of the welds, and the overall trends for residual stresses measured from the start and end parts of the welds were also identical to those measured from the center part of the welds.

## Data availability

The datasets generated and/or analyzed during the current study are available from the corresponding author on request.

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

## Acknowledgements
This work was carried out based on POSCO's PosZET® GIGA. J. Moon wishes to thank the POSCO technical research laboratory for technical and financial support. Finally, the authors also thank Dr. Dierk Raabe in Max-Planck-Institut für Eisenforschung for his advice on this investigation.

## Author contributions
J.M. and G.B. carried out microstructure characterization and wrote the first version of the manuscript. G.B. designed the chemical compositions of the welds, prepared the welded samples and carried out the residual stress measurement and the Charpy V-notch impact and fatigue tests. C.S. carried out the microtensile tests. M.J.K. prepared the samples for microstructure characterization and carried out the SEM and EBSD observations. D.I.K. and D.J.C. performed the in situ EBSD tensile tests. B.H.L. analyzed Nb segregation along the prior austenite grain boundaries. C.H.L. supported the dilatometer experiments for establishing the CCT diagram in the welds. H.U.H., D.W.S., D.P., and B.Y.J. discussed the experimental results and commented on the manuscript.

## Competing interests
POSCO has filed a patent, PCT/KR2023/018810 (inventor: G.B.) that covers the PosZET® GIGA's chemistry and methods reported in this article. The authors declare no other competing interests.
