## [Peer Review File · Nature Communications]

Ultrastrong and ductile steel welds achieved by fine interlocking microstructures with film-like retained austeniteREVIEWER COMMENTS

Reviewer #1 (Remarks to the Author):

The paper presents an interesting approach to improving the weld quality of ultra-high strength steels by substituting Ni with Nb and Cr. It is a novel approach that combines well known facts and experience from the field of welding to overcome the microstructure coarsening and the resulting mechanical property degradation that comes with welding of AHSS. The microstructure design concept and execution are not significantly novel and does not bring novelty from a fundamental point of view, but a successful technological achievement.

Several aspects of the microstructure are analysed and designed in this work, complemented with thermodynamic calculations. The desired microstructural design in this work gets inspiration from the so-called superbainite or nanobainite, known to yield superior mechanical properties by combining very small platelets of bainitic ferrite and retained austenite. However, the properties that superbainitic microstructures exhibit are radically different (exceeding ultimate tensile stress of 2GPa), which is not what is seen here. A careful distinction needs to be made in this point about the relation of the studied microstructures with the nano-bainitic microstructures from literature.

Additionally, other favourable microstructure constituents are introduced, i.e. acicular ferrite which nucleates on oxides, which is standard practice for HSLA weld material design.

The paper is well structured and a combination of state of the art material characterisation techniques are used. The methodology is sound and it follows the norms in the field. The experimental procedure is well described and makes the experiments reproducible.

However, the following points would require further elaboration:

1. The authors substitute Nickel with Cr and Nb in the alloy design, claiming that this concept is cheaper and more sustainable. The sustainability elaboration is not justified in the text and the sources for indicated prices per ton of alloy produced are also not stated. Additionally, Nb is very high in the list of critical raw materials for the EU (EU Report 2017), see figure 1 below:

Figure 1: criticality assessment for various raw materials, reproduced from: <https://h2020-tarantula.eu/why-tungsten-niobium-and-tantalum-are-critical-raw-materials/>

According to EU policy, Nb is a critical raw material and it should be avoided in new designs if possible. Nickel remains a very low supply risk material, even though it is becoming increasingly expensive. How do the authors see the potential influence of the scarcity of Nb to the indicated prices?

2. The tensile tests were performed in miniature samples, while fatigue tests were performed on standard samples. How do the authors comment on the influence of specimen size on the reported mechanical properties? How do their materials compare with other AHSS welding consumables in the market today?

3. The fraction of retained austenite is very small (<2%) if compared to nanobainitic microstructures (8-15% or more). Bhadeshia in Nanostructured bainite (2009)

<https://doi.org/10.1098/rspa.2009.0407> shows microstructures with at least 10% of retained austenite to benefit from this metastable phase. It can be argued that a fraction of 2% of retained austenite in the microstructure, its effect is overstated in the present work.

4. The residual stress measurements indicate a significant difference between the two steel alloy designs. The method used only examines the surface of the weld, and the measured effect is likely to represent only the conditions there and not at the bulk. Alternative methods would be more appropriate to conclude on the residual stress status of these welds (E.g. hole drilling, neutron diffraction). Additionally, the effect of the tetragonality of the bainitic ferrite lattice on the residual stress is highly speculative and does not justify the residual stress formation directly.

Reviewer #2 (Remarks to the Author):

Review Comments:

Steels are the most widely used structural materials. The strengthening and toughening of the welding

joint is a long-standing challenge. Demands for the safety of structures and the weight reduction of structural parts are ever increasing. The present work provide some interesting results on ultrastrong and ductile steel welds achieved by fine interlocking microstructures with film-like retained austenite. It is found that a new alloying design with Nb and Cr creates ultrastrong and ductile steel welds with enhanced tensile properties and impact toughness, and fatigue strength by forming the finer interlocking microstructure and the higher fraction of film-like retained austenite, in the meantime, with a much lower material costs. The results are very helpful for design of low-alloy ultra-high strength structural steels.

However, the innovation of this manuscript is insufficient for nature communications. Meanwhile, the results and discussion are not fully clear to reveal the microstructure effects on scientific bases.

(1) The mixed acicular ferrite, bainite, martensite and retained austenite microstructure is obtained in the welds of ultra-high strength steels by utilizing Nb and Cr instead of Ni, how can authors distinguish between the three types microstructure of acicular ferrite, bainite, and martensite? And what are their specific volume fractions?

(2) The fraction of retained austenite is very low in the welds in Extended Data Table 1. What experimental instrument was used to measure? and what is the measurement accuracy or deviation? At the same time, the stability of retained austenite is also closely related to its carbon content, and the manuscript did not address the impact of the carbon content of retained austenite.

(3) It is generally believed that the excellent plasticity and toughness are mainly attributed to the induced plastic martensitic transformation (TRIP effect) of metastable retained austenite during the strain process. However, there are significant differences in the carbon content, volume fraction, size, morphology, and microstructure distribution of different retained austenite (for example block and film), and the TRIP effect of different retained austenite is not discussed in the manuscript.

(4) Fig. 6b shows the SEM micrograph of the fractured surface in the 1.0 GPa H-Nb steel weld after the impact test at -40°C. However, there is a lack of comparative sample morphology (1.0 GPa H-Ni steel weld). At the same time, Figures 6a and 6c only have data for 1.0 GPa H-Nb and 1.0 GPa H-Ni steel weld, but no data for 1.2 GPa H-Nb and 1.2 GPa H-Ni steel weld. What is the reason?

(5) The weld compositions are influenced by base metal, welding wire and welding method. The base metal, welding wire and welding method should be specified.

(6) In line 99, it is stated that in the text "In addition, both tensile strengths and uniform elongations in the welds of UHSS increase by 14 MPa and 0.8% (for 1.0 GPa-grade UHSS welds) and 86 MPa and 0.5% (for 1.2 GPa-grade UHSS welds). The impact toughness and fatigue strength are also improved significantly." It is better to specify the impact toughness and fatigue strength improvements.

(7) In line 56, it is stated that in the text "In Particular, various kinds of UHSS such as transformation-induced plasticity (TRIP) steel", "In Particular" should be "In particular".

(8) In line 56, it is stated that in the text "a new weld composition design leading to superior uniform elongation. In Figs. 5(c) and (d)", the stain values should be "the strain values".

Reviewer #3 (Remarks to the Author):

Accepted with Minor corrections

Overall: Overall an original research paper, systematically demonstrating how the chemical composition of the resulting welded microstructure can achieve ultra strong and ductile material, through fine interlocking microstructures with film-like retained austenite. The methodology is sound and the conclusions reasonable.

Advanced characterisation & analysis techniques have been employed, including SEM, TEM microscopy, FIB, EBSD, APT, SADP, micro-tensile tests, in-situ EBSD tensile tests, Charpy V-notch impact tests, fatigue tests, and X-ray diffraction residual stress measurements.

The figures are carefully constructed and original. The schematics demonstrating the hierarchical microstructures in the weld are very explanatory and impactful.

Points for consideration:

- Abstract has to be ~150 words

- Paragraph 'Sample preparation':

- Line 384: '...thin plates..' - what is the thickness of the plates? Is it 2.9 mm? Then consider transferring lines 445-446 in paragraph 'sample preparation'.
- Would it be possible to mention the welding parameters used? Perhaps include a table.
- Furthermore, it would be useful to state the plates geometrical characteristics. The easiest way to demonstrate that would be adding dimensions eg in Fig 1(a); dimensions to include length, width, thickness, overlap extent.

- Line 529: This table is highlighting the cases that are considered for drawing conclusions, hence I propose treating amending the table caption to 'Table 2' and the description stating that the table includes the wt% of the elements of interest.

- Fig. 1: re-arrange the sub-figures, for (d), (e), (c) to be in orders (perhaps flip e with d)

- Fig. 3: the distance is measured from where? Please indicate either the fusion zone, the weld centre line or some reference

- Extended Data Figure 3. Residual stress distribution around the welds.: The reference for the horizontal axis 'Distance from the weld, mm' is not clear.

- Is a macrograph of the welds transverse cross-section available? If so, it could be added to one of the figures (eg extended fig 2, extended fig 3, or fig 3) to demonstrate the extent of the metallurgical zones, where the residual measurements were taken from, where the PAGB interface lies with respect to the macroscopic characteristics of the weld.

Response letter

Manuscript number: NCOMMS-23-26635A-Z

Title: Ultrastrong and ductile steel welds achieved by fine interlocking microstructures with film-like retained austenite

(Details response to the reviewers' comments)

Reviewer #1

	Reviewer's comments (Q) and authors' answer (A)
	(Q) The paper presents an interesting approach to improving the weld quality of ultra-high strength steels by substituting Ni with Nb and Cr. It is a novel approach that combines well known facts and experience from the field of welding to overcome the microstructure coarsening and the resulting mechanical property degradation that comes with welding of AHSS. The microstructure design concept and execution are not significantly novel and does not bring novelty from a fundamental point of view, but a successful technological achievement. Several aspects of the microstructure are analysed and designed in this work, complemented with thermodynamic calculations. The desired microstructural design in this work gets inspiration from the so-called superbainite or nanobainite, known to yield superior mechanical properties by combining very small platelets of bainitic ferrite and retained austenite. However, the properties that superbainitic microstructures exhibit are radically different (exceeding ultimate tensile stress of 2GPa), which is not what is seen here. A careful distinction needs to be made in this point about the relation of the studied microstructures with the nano-bainitic microstructures from literature. Additionally, other favourable microstructure constituents are introduced, i.e. acicular ferrite which nucleates on oxides, which is standard practice for HSLA weld material design. The paper is well structured and a combination of state of the art material characterisation techniques are used. The methodology is sound and it follows the norms in the field. The experimental procedure is well described and makes the experiments reproducible. → (A) Thank you for your kind and positive comments.
1	(Q) The authors substitute Nickel with Cr and Nb in the alloy design, claiming that this concept is cheaper and more sustainable. The sustainability elaboration is not justified in

	the text and the sources for indicated prices per ton of alloy produced are also not stated. Additionally, Nb is very high in the list of critical raw materials for the EU (EU Report 2017), see figure 1 below: Figure 1: criticality assessment for various raw materials, reproduced from: https://h2020-tarantula.eu/why-tungsten-niobium-and-tantalum-are-critical-raw-materials/ According to EU policy, Nb is a critical raw material and it should be avoided in new designs if possible. Nickel remains a very low supply risk material, even though it is becoming increasingly expensive. How do the authors see the potential influence of the scarcity of Nb to the indicated prices? → (A) Thank you for your valuable comments. I read the EU report (2017) you mentioned carefully. I understand your concern about the potential risk of supply of Nb because its production and supply are almost dependent on sourcing from Brazil. However, when considering the amount of Nb added in our new alloy design and its benefits, I think we can take this potential supply risk of Nb. We removed 1.0-1.5wt.% Ni and then just added very small amounts of Nb (0.02-0.03wt.%). In addition, the recycling rate of Nb is very high (Please see the paper published in Science: Barbara K. Reck and T. E. Graedel, Challenges in Metal Recycling, Science 337 (2012) 690). I believe these facts can possibly minimize the impact in the situation of supply risk. Meanwhile, in this investigation, we focused on the advantage of lower material cost obtained by replacing high amounts of Ni (1.0~1.5wt.%) with low amounts of Nb (0.02~0.03wt.%) and Cr (0.3~0.35wt.%), as shown in Table 1. As confirmed in this study, this alloy design strategy not only reduced the material cost greatly but also improved mechanical properties by forming the finer interlocking microstructure with higher fraction of retained austenite.
2	(Q) The tensile tests were performed in miniature samples, while fatigue tests were performed on standard samples. How do the authors comment on the influence of specimen size on the reported mechanical properties? How do their materials compare with other AHSS welding consumables in the market today? → (A) As you correctly point out, specimen size is indeed a critical factor in determining the mechanical properties of materials. It is well-established in the field of materials science and applied mechanics that the mechanical properties of materials can be substantially affected by specimen size, particularly when the specimen dimensions approach or fall below the characteristic length scales of the microstructure. This phenomenon, known as size effect, has been extensively studied and is a crucial

	consideration when characterizing the mechanical behavior of materials. One of the prominent manifestations of the size effect is the reduction in yield strength with decreasing sample size, especially in polycrystalline metallic materials. This reduction in yield strength is often attributed to the decreasing proportion of grain boundaries (GBs) as the specimen size decreases. Grain boundaries are known to impede dislocation motion, and with fewer GBs present, the resistance to dislocation motion diminishes, resulting in a decrease in yield strength. Therefore, it is imperative that specimen dimensions are sufficiently larger than microstructural constraints to obtain meaningful yield strength measurements that represent the bulk material's behavior. From a practical standpoint, drawing from the data reported in the literature [1-4], it is recommended that the ratio t/d (where t represents the thickness of the flat tensile specimen and d signifies the grain size) should exceed 10 to guarantee that the measured mechanical properties are representative of the material's bulk behavior. Recent theoretical study [5] have shown that at the values greater than 15, the yield strength becomes size-independent and converges toward the bulk yield strength. In our study, the value of t/d is 113-185, significantly exceeding the proposed critical threshold of 10 or 15. As a result, the mechanical properties derived from the sub-size tensile specimens utilized in our research can be confidently regarded as equivalent to the bulk material's properties. We appreciate the opportunity to address this important aspect of our study. I mention this explanation in the revision. [1] Miyazaki S, Shibata K, Fujita H, Acta Metal 1979;27;855 [2] Igata N, Miyahara K, Ohno K, Uda T, J Nucl Mater 1984;122&123;354 [3] Miyahara K, Tada C, Uda T, Igata N, J Nucl Mater 1985;133&134;506 [4] Keller C, Hug E, Retoux R, Feaugas X, Mech Mater 2010;42;44 [5] Liu W, Liu Y, Cheng Y, Chen L, Yu L, Yi X, Duan H, Phys Rev Lett 2020;124;235501
3	(Q) The fraction of retained austenite is very small (<2%) if compared to nanobainitic microstructures (8-15% or more). Bhadeshia in Nanostructured bainite (2009) https://doi.org/10.1098/rspa.2009.0407 shows microstructures with at least 10% of retained austenite to benefit from this metastable phase. It can be argued that a fraction of 2% of retained austenite in the microstructure, its effect is overstated in the present work. → (A) Thank you for your advice. I understand your concerns about the effect of retained austenite with small fraction (2%) on ductility improvement in H-Nb steel welds.

	Previous studies [1-2] reported that a decrease in the fraction of retained austenite increases carbon content in retained austenite and thereby makes retained austenite more stable, finally making retained austenite difficult to be transformed to martensite during deformation. That is, this indicates that the effect of retained austenite on ductility improvement might be reduced by decreasing its fraction. However, we showed clear evidence of the retained austenite to martensite transformation in welds during tensile deformation (Fig. 5(e)) even though the fraction of retained austenite in welds was not as high as those in typical TRIP steels or nanostructured bainitic steel. Figure 5(e) indicates that the retained austenite distributed in welds is gradually transformed to martensite with straining, contributing to strain hardening and improvement of ductility. We appreciate the opportunity to explain this important point of the role of retained austenite in welds. We mention this explanation in the revision. [1] X.X. Dong et al., Mater. Sci. & Eng. A 2022;833;142580 [2] K. Sugimoto et al., ISIJ Int. 1992;32;1311
4	(Q) The residual stress measurements indicate a significant difference between the two steel alloy designs. The method used only examines the surface of the weld, and the measured effect is likely to represent only the conditions there and not at the bulk. Alternative methods would be more appropriate to conclude on the residual stress status of these welds (E.g. hole drilling, neutron diffraction). → (A) Thank you for your advice. We measured the residual stress of the welds through X-ray diffractometer testing using a XSTRESS 3000 machine. As your comment, this method can only examine the residual stress on the surface. However, in this investigation, we thought that the residual stress values measured on the surface may represent the residual stress of the weld because a thickness of welded specimen is very thin (2.9 mm). According to previous work [1], the residual stress deviation between the surface and the center position decreases significantly as the thickness decreases. [1] B. Qiang, Y. Li, C. Yao, X. Wang, Y. Gu, J. Mater. Process. Technol. 2018; 251; 54 (Q) Additionally, the effect of the tetragonality of the bainitic ferrite lattice on the residual stress is highly speculative and does not justify the residual stress formation directly. → (A) Thank you for your very useful comment. Yes, you are right. We discussed a possible effect of tetragonality of the bainitic ferrite lattice on the residual stress. As your

	comment, I am sorry that we didn't show direct evidence of the relationship between tetragonality and residual stress in this investigation. However, the previous work [1] reported that a tetragonality generated during austenite to martensite transformation induces a compressive internal residual stress. Therefore, we think that different results of residual stress measured in each weld, as shown in Extended Data Figure 3, might be originated from difference values between tetragonality of the bainite in H-Ni/L-Nb and H-Nb/L-Ni welds, so we explained about different residual stress behaviors between H-Ni/L-Nb and H-Nb/L-Ni welds with the effect of different tetragonality of the bainitic ferrite lattice caused by different transformation temperature during cooling in this work. To verify experimentally what you point out, we are preparing further experiments for measuring tetragonality value of the bainitic ferrite lattice in welds and analyzing the relationship between tetragonality and residual stress. We will report these results in next paper. Please understand that the present work mainly focuses on the correlation between microstructure changes and mechanical properties (tensile, impact, and fatigue properties) in UHSS welds by adding Nb and Cr instead of Ni. [1] D. Fukui, N. Nakada, S. Onaka, Acta Mater. 2020; 196; 660
--	---

Reviewer #2

	Reviewer's comments (Q) and authors' answer (A)
-	(Q) Steels are the most widely used structural materials. The strengthening and toughening of the welding joint is a long-standing challenge. Demands for the safety of structures and the weight reduction of structural parts are ever increasing. The present work provide some interesting results on ultrastrong and ductile steel welds achieved by fine interlocking microstructures with film-like retained austenite. It is found that a new alloying design with Nb and Cr creates ultrastrong and ductile steel welds with enhanced tensile properties and impact toughness, and fatigue strength by forming the finer interlocking microstructure and the higher fraction of film-like retained austenite, in the meantime, with a much lower material costs. The results are very helpful for design of low-alloy ultra-high strength structural steels. However, the innovation of this manuscript is insufficient for nature communications. Meanwhile, the results and discussion are not fully clear to reveal the microstructure effects on scientific bases. → (A) Thank you for your valuable comment. In this study, the authors have investigated new weld microstructure for GPa-grade automotive ultra-high strength steels. Compared

	to conventional material, we added Nb and Cr instead of Ni in welds. By this alloy design, we not only reduced the materials costs by approximately 45% but also achieved much superior mechanical properties in welds such as tensile strength, uniform elongation, impact toughness, and fatigue strength, as compared to conventional welds. Moreover, the authors revealed the mechanisms for improving mechanical properties in welds by adding Nb and Cr instead of Ni, through thermodynamic calculations, intensive microstructure analyses (TEM, TEM-EBSD, APT), micro-tensile tests, in-situ EBSD tensile tests, and dilatometer experiments. We believe that our study on the new design of weld compositions and microstructures could be a potential strategy to replace conventional ones for welding of UHSS owing to its much superior mechanical properties with cheaper material cost. We believe that this paper will be of interest to the readership of Nature Communications because our strategy, i.e., the cheaper alloy design but strong and ductile microstructure achieved, may inspire similar research efforts into the development of other ultrastrong structural materials. This design strategy is not limited to UHSS welds but applicable to any structural material.
1	(Q) The mixed acicular ferrite, bainite, martensite and retained austenite microstructure is obtained in the welds of ultra-high strength steels by utilizing Nb and Cr instead of Ni, how can authors distinguish between the three types microstructure of acicular ferrite, bainite, and martensite? And what are their specific volume fractions? → (A) This is very important question. We appreciate the opportunity to explain this important point of microstructures in welds. Usually, the acicular ferrite nucleates on inclusions and grows radially, as shown in Fig. 1(b). The distinction between the bainite and the martensite is very difficult because both of them have same morphology of lath-shape, as shown in 4(c). The most significant difference between the bainite and the martensite is a distribution of carbon atoms. In the case of martensite, C atoms are supersaturated within the martensite laths. In the case of bainite, C atoms form the cementite within the lath (lower bainite) and along the lath boundaries (upper bainite). In this investigation, we found carbide-free bainite in welds and then distinguish it by the distribution of retained austenite along lath boundaries, i.e., the lath structures with retained austenite has been regarded as the bainite, as shown in Fig. 4(c). Next, the measurement of specific volume fractions of acicular ferrite, bainite, and martensite is very difficult to obtain their exact values due to their similar lath shape and same BCC based crystal structure. Nevertheless, many researchers have distinguished them and measured their fractions using the Misorientation distribution

data obtained from EBSD analyses. Please see the paper : J.C.F. Jorge et al., J. Mater. Res. Technol., 2021; 10; 471-501. In general, the bainite has a very high fraction of low-angle grain boundaries with a strong peak in Misorientation profile below 10 °. Next, Acicular ferrite has a higher fraction of high angle grain boundaries with Misorientation angle greater than 47 °. Finally, the martensite has a higher fraction of boundaries with low misorientation angle in the 2.5 – 8 ° range. The figure below shows an example of Misorientation angle distribution in 1.0GPa H-Nb/L-Ni steel weld measured by EBSD analyses. The displays indicates that the fraction of Misorientation angle above 47 ° (acicular ferrite fraction) is 55 % in H-Nb/L-Ni steel weld, indicating that the fraction of bainite and martensite is approximately 45 % including 0.3-1.8% retained austenite. However, I am sorry that it is very difficult to measure exact fractions of each phase so that I don't state this issue in the manuscript. More important microstructural differences between conventional and developed welds affecting weld mechanical properties in our work are the effective grain size and fraction of retained austenite, as shown in Fig. 2(i).

Fig. Misorientation angle distribution measured in 1.0GPa H-Nb/L-Ni weld

2

(Q) The fraction of retained austenite is very low in the welds in Extended Data Table 1. What experimental instrument was used to measure? and what is the measurement accuracy or deviation?

→ (A) Thank you for your question. Here we measured the fraction of retained austenite through EBDS phase map analyses. I mention this statement in the revision.

This way to measure the fraction of retained austenite using EBSD phase map has been recognized by many previous researchers as a highly reliable method as follow references:

[1] F. Gao et al., J. Mater. Res. Technol., 2022;20;1976.
 [2] S. Pashangeh et al., Metals, 2019;9(5);492.
 [3] B. Sun et al., Nature Mater., 2021;20;1629.

(Q) At the same time, the stability of retained austenite is also closely related to its carbon

	content, and the manuscript did not address the impact of the carbon content of retained austenite. → (A) Thank you for your advice. Previous studies [1-2] reported that a decrease in the fraction of retained austenite increases carbon content in retained austenite and thereby makes retained austenite more stable, finally making retained austenite difficult to be transformed to martensite during deformation. That is, this indicates that the effect of retained austenite on ductility improvement might be reduced by decreasing its fraction. As you mention, the fraction of retained austenite in the welds measured in this study was quite low, indicating that the retained austenite to martensite transformation in the welds during deformation might not occur well. However, we presented clear evidence of the retained austenite to martensite transformation in welds during tensile deformation (Fig. 5(e)) even though the fraction of retained austenite in welds was not as high as those in typical TRIP steels or nanostructured bainitic steel. Figure 5(e) indicates that the retained austenite distributed in welds is gradually transformed to martensite with straining, contributing to strain hardening and improvement of ductility. We appreciate the opportunity to explain this important point of the role of retained austenite in welds. I mention this explanation in the revision. [1] X.X. Dong et al., Mater. Sci. & Eng. A 2022;833;142580 [2] K. Sugimoto et al., ISIJ Int. 1992;32;1311
3	(Q) It is generally believed that the excellent plasticity and toughness are mainly attributed to the induced plastic martensitic transformation (TRIP effect) of metastable retained austenite during the strain process. However, there are significant differences in the carbon content, volume fraction, size, morphology, and microstructure distribution of different retained austenite (for example block and film), and the TRIP effect of different retained austenite is not discussed in the manuscript. → (A) Yes, I agree with your opinion. The morphology, volume fraction, carbon content, and distributed position of retained austenite are very important factors determining the effect of retained austenite on the TRIP effect. We added the discussion on these in the revision.
4	(Q) Fig. 6b shows the SEM micrograph of the fractured surface in the 1.0 GPa H-Nb steel weld after the impact test at - 40°C. However, there is a lack of comparative sample morphology (1.0 GPa H-Ni steel weld). → (A) We add its SEM micrograph of the fractured surface in 1.0 GPa H-Ni steel weld. Please see the image added in Fig. 6.

	(Q) At the same time, Figures 6a and 6c only have data for 1.0 GPa H-Nb and 1.0 GPa H-Ni steel weld, but no data for 1.2 GPa H-Nb and 1.2 GPa H-Ni steel weld. What is the reason ? → (A) For the impact testing, the thickness of the sample should be at least 2.5 mm for making sub-size impact test specimen. However, 1.2 GPa UHSS plate welded in this investigation is a cold-rolled TRIP steel with a thickness less than 2.0 mm and thus we couldn't perform the impact test for 1.2 GPa steel welds. We are sorry that we can't show the impact test data of 1.2 GPa steel welds, but we believe that the impact test data of Fig. 6(a) shows clearly a significant improvement of impact properties in new steel weld (H-Nb/L-Ni weld) compared to conventional steel weld (H-Ni/L-Nb weld).
5	(Q) The weld compositions are influenced by base metal, welding wire and welding method. The base metal, welding wire and welding method should be specified. → (A) We add detailed welding parameters in Extended Data Table 1. And the weld metal compositions are given in Table 1. Cold-rolled TRIP steel (1.2 GPa-grade UHSS) and Hot-rolled Complex Phase (CP) steel (1.0 GPa-grade UHSS) were selected as the base metals for this investigation. This investigation aimed to improve weld quality by controlling weld microstructure caused by changing weld metal chemical composition. Therefore, we think that it is enough to show the compositions of the welds to understand the relationship between weld compositions, microstructure evolution, and mechanical properties. We are truly sorry that we can't disclose the base metal and welding wire compositions at this point. Dr. G. Bae, one of the co-author, designed the chemical compositions of the welding wires and prepared the welded samples using the base metals (1.2 GPa TRIP steel & 1.0 GPa CP steel) developed by his company. Please understand that it is difficult for Dr. G Bae to disclose the base metal and welding wire composition at this point due to his company's research security policy.
6	(Q) In line 99, it is stated that in the text "In addition, both tensile strengths and uniform elongations in the welds of UHSS increase by 14 MPa and 0.8% (for 1.0 GPa-grade UHSS welds) and 86 MPa and 0.5% (for 1.2 GPa-grade UHSS welds). The impact toughness and fatigue strength are also improved significantly." It is better to specify the impact toughness and fatigue strength improvements. → (A) Thank you for your advice. As your comment, we specify the improvements of impact and fatigue properties.
7	(Q) In line 56, it is stated that in the text "In Particular, various kinds of UHSS such as transformation-induced plasticity (TRIP) steel", "In Particular" should be "In particular". → (A) Thank you for your indication. We correct the typo.

8	(Q) In line 56, it is stated that in the text “a new weld composition design leading to superior uniform elongation. In Figs. 5(c) and (d)”, the stain values should be “the strain values”. → (A) Thank you for your indication. We correct the typo.
---	--

Reviewer #3

Reviewer’s comments (Q) and authors’ answer (A)	
1	(Q) Accepted with Minor corrections Overall: Overall an original research paper, systematically demonstrating how the chemical composition of the resulting welded microstructure can achieve ultra strong and ductile material, through fine interlocking microstructures with film-like retained austenite. The methodology is sound and the conclusions reasonable. Advanced characterisation & analysis techniques have been employed, including SEM, TEM microscopy, FIB, EBSD, APT, SADP, micro-tensile tests, in-situ EBSD tensile tests, Charpy V-notch impact tests, fatigue tests, and X-ray diffraction residual stress measurements. The figures are carefully constructed and original. The schematics demonstrating the hierarchical microstructures in the weld are very explanatory and impactful. → (A) Thank you for your kind and positive comments.
1	(Q) Abstract has to be ~150 words. → (A) Thank you for your comment. We reduce the length of abstract to less than 150 words.
2	(Q) Paragraph 'Sample preparation':  • Line 384: '...thin plates..' - what is the thickness of the plates? Is it 2.9 mm? Then consider transferring lines 445-446 in paragraph 'sample preparation'. → (A) I am sorry for making you confused. We used the plates with different thickness for lap-joint welding and butt-joint welding. In the case of lap-joint welding, we prepared thin plates with thickness of 2.0 mm (1.0 GPa grade steel) and 1.6 mm (1.2GPa grade steel) for evaluating microstructures and fatigue strengths of the welds. In the case of butt-welding, we used a plate with thickness of 2.9 mm for measuring residual stresses and impact toughness of the welds.  • Would it be possible to mention the welding parameters used? Perhaps include a table. → (A) Thank you for your comment. We add a table showing the welding parameters used in this work in Extended Data Table 1.  • Furthermore, it would be useful to state the plates geometrical characteristics. The easiest way to demonstrate that would be adding dimensions eg in Fig 1(a); dimensions to include length, width, thickness, overlap extent. → (A) As your comment, we write a dimension of the welded plate in the section of ‘Sample

	preparation' of 'Methods'.
3	(Q) Line 529: This table is highlighting the cases that are considered for drawing conclusions, hence I propose treating amending the table caption to 'Table 2' and the description stating that the table includes the wt% of the elements of interest. → (A) Thank you for your kind comments. I revise the manuscript based on your comments.
4	(Q) Fig. 1: re-arrange the sub-figures, for (d), (e), (c) to be in orders (perhaps flip e with d). → (A) Thank you for your advice. As your comment, we rearrange the sub-figures of Fig. 1.
5	(Q) Fig. 3: the distance is measured from where? Please indicate either the fusion zone, the weld centre line or some reference → (A) As shown in Fig. 3, we analyzed Nb and C segregation along the prior austenite grain boundary (PAGB) in the weld centerline. In Fig. 3(d), we measured C and Nb contents across the PAGB interface (along the direction marked by blue arrows in Fig. 3(b, c). Distance '0' in Fig. 3(d) indicates the point the PAGB starts while following in the direction of the blue arrow. I mention this explanation in the caption of Fig. 3.
6	(Q) Extended Data Figure 3. Residual stress distribution around the welds.: The reference for the horizontal axis 'Distance from the weld, mm' is not clear. → (A) I am sorry for making you confused. I add schematic diagram showing the points that the residual stress was measured in Extended Data Fig. 3(a). 'Distance from the weld, mm' in Extended Data Fig. 3(b) indicates the distance from the fusion boundary in the weld. I believe that a schematic diagram added in Extended Data Fig. 3(a) will help you understand the experiment for residual stress measurement.
7	(Q) Is a macrograph of the welds transverse cross-section available? If so, it could be added to one of the figures (eg extended fig 2, extended fig 3, or fig 3) to demonstrate the extent of the metallurgical zones, where the residual measurements were taken from, where the PAGB interface lies with respect to the macroscopic characteristics of the weld. → (A) Unfortunately, we don't have macrographic cross-sectional images of butt welds prepared for residual stress measurement. Instead, in order to help you understand the residual stress measurement positions, we add Extended Data Fig. 3(a) in the revision.

(Additional important revised point in the revision)

In this revised version, Figures 5(c) and (d) now depict strain hardening rates derived directly from experimental data. In the initial manuscript, the true stress-strain and strain hardening rates curves were generated through nonlinear fitting of experimental data, utilizing the empirical Hollomon constitutive stress-strain relation. However, to enhance clarity and eliminate potential ambiguity associated with

selecting a specific constitutive model from various options, we have opted to replace the original graphs. The current representation solely relies on experimental data, offering a more model-independent and transparent depiction of the stress-strain behavior in our study. We believe this modification strengthens the reliability and objectivity of our findings.

REVIEWERS' COMMENTS

Reviewer #1 (Remarks to the Author):

The previous comments have been adequately incorporated in the new manuscript.

Reviewer #2 (Remarks to the Author):

Steel is the most widely used structural material. This paper raises some interesting results for the study of ultra-strong and ductile steel welds. A new design concept for Nb and Cr alloy design, which results in a finer interlocking microstructure and a higher proportion of film-like residual austenite, allows for the fabrication of ultra-strong and ductile steel welds with enhanced tensile properties, impact toughness, and fatigue strength, and at the same time, at a lower cost of material. It is a successful technical achievement. The author has carefully revised the manuscript. However, there are still some minor questions that need to be considered.

1.Line 149-151:"and the fraction of film-like retained austenite increased with the addition of small amounts of Nb and Cr by replacing Ni."-Fig.2i summarizes the results of the content of retained austenite, please also describes the content changes in the text.

2.Line 167-172: The logicity of the sentence needs to be revised.

Reviewer #3 (Remarks to the Author):

The authors have diligently addressed all the concerns and queries raised during the review process. The revisions made have strengthened the quality and clarity of the manuscript. The authors' positive response and willingness in incorporating the suggestions and making amendments, is much appreciated. I believe that the manuscript is overall rigor and will make a valuable contribution to the field.

Response letter

Manuscript number: NCOMMS-23-26635B

Title: Ultrastrong and ductile steel welds achieved by fine interlocking microstructures with film-like retained austenite

(Details response to the reviewers' comments)

Reviewer #1

	Reviewer's comments (Q) and authors' answer (A)
	(Q) The previous comments have been adequately incorporated in the new manuscript. → (A) Thank you for your kind and positive comments.

Reviewer #2

	Reviewer's comments (Q) and authors' answer (A)
-	(Q) Steel is the most widely used structural material. This paper raises some interesting results for the study of ultra-strong and ductile steel welds. A new design concept for Nb and Cr alloy design, which results in a finer interlocking microstructure and a higher proportion of film-like residual austenite, allows for the fabrication of ultra-strong and ductile steel welds with enhanced tensile properties, impact toughness, and fatigue strength, and at the same time, at a lower cost of material. It is a successful technical achievement. The author has carefully revised the manuscript. However, there are still some minor questions that need to be considered. → (A) Thank you for your kind comments. We revise the manuscript based on your minor comments.
1	(Q) Line 149-151: "and the fraction of film-like retained austenite increased with the addition of small amounts of Nb and Cr by replacing Ni."-Fig.2i summarizes the results of the content of retained austenite, please also describes the content changes in the text. → (A) Thank you for your valuable comment. As your comment, I mention the changes of retained austenite fraction caused by the addition of Nb and Cr instead of Ni in the text.
2	(Q) Line 167-172: The logicity of the sentence needs to be revised. → (A) Thank you for your advice. I agree with your comment. In the revision, we revise these sentences more logically.

Reviewer #3

	Reviewer's comments (Q) and authors' answer (A)
	(Q) The authors have diligently addressed all the concerns and queries raised during the review process. The revisions made have strengthened the quality and clarity of the manuscript. The authors' positive response and willingness in incorporating the suggestions and making amendments, is much appreciated. I believe that the manuscript is overall rigor and will make a valuable contribution to the field. → (A) Thank you for your kind and positive comments.